# MULTI-SCALE ATTRIBUTED NODE EMBEDDING

## ABSTRACT

We present network embedding algorithms that capture information about a node from the local distribution over node attributes around it, as observed over random walks following an approach similar to Skip-gram. Observations from neighborhoods of different sizes are either pooled (AE) or encoded distinctly in a *multi-scale* approach (MUSAE). Capturing attribute-neighborhood relationships over multiple scales is useful for a diverse range of applications, including latent feature identification across disconnected networks with similar attributes. We prove theoretically that matrices of node-feature pointwise mutual information are implicitly factorized by the embeddings. Experiments show that our algorithms are robust, computationally efficient and outperform comparable models on social, web and citation network datasets.

## 1 INTRODUCTION

Node embedding is a fundamental technique in network analysis that serves as a precursor to numerous downstream machine learning and optimisation tasks, e.g. community detection, network visualization and link prediction (Perozzi et al., 2014; Grover & Leskovec, 2016; Tang et al., 2015). Several recent network embedding methods, such as *Deepwalk* (Perozzi et al., 2014), *Node2Vec* (Grover & Leskovec, 2016) and *Walklets* (Perozzi et al., 2017), achieve impressive performance by learning the network structure following an approach similar to *Word2Vec Skip-gram* (Mikolov et al., 2013b), originally designed for word embedding. In these works, sequences of neighboring nodes are generated from random walks over a network, and representations are distilled from extracted node-node proximity statistics that capture local neighbourhood information.

When the nodes of a network have attributes (or features), their embeddings can be used to capture information about the attributes in their local neighbourhood. For a social network, attributes might represent a person's interests, habits, history or preferences. The pattern of node attributes are often similar in a neighborhood, and conversely, nodes with similar attributes are more likely to be connected. This property is known as *homophily*. Attributed network embedding methods (Yang et al., 2015; Huang et al., 2017; Liao et al., 2018) leverage this additional information to supplement that of node neighbourhood structure, benefiting many applications, e.g. recommender systems, node classification and link prediction (Yang et al., 2018; Yang & Yang, 2018; Zhang et al., 2018).

The neighborhood of a node can be considered at different path lengths, or *scales*. In a social network, near neighbors may correspond to classmates, whereas nodes separated by greater scales may be in different cities or countries. Attributes of neighbors at different scales can be considered separately (*multi-scale*) or *pooled* in some way (e.g. weighted average). Figure 1a shows how the attribute distribution over neighbourhoods at different scales can indicate nodes with similar network *roles* even if they are distant in the network, or even in different networks. Methods that take attributes of nearby nodes into account generalizes those that do not, e.g. Perozzi et al. (2017), for which feature vectors can be considered standard basis vectors.

Many embedding methods correspond to matrix factorization, indeed some attributed embedding methods (e.g. Yang et al. (2018)) explicitly factorize a matrix of link-attribute information. Embeddings learned using Skip-gram are known to factorize a matrix of *pointwise mutual information* (PMI) of co-occurrences between each word and local *context* words (Levy & Goldberg, 2014). Related network embedding methods (Perozzi et al., 2014; Grover & Leskovec, 2016; Tang et al., 2015; Qiu et al., 2018) also implicitly factorize PMI matrices based on the probability of encountering each (context) node on a random walk from each starting node (Qiu et al., 2018).

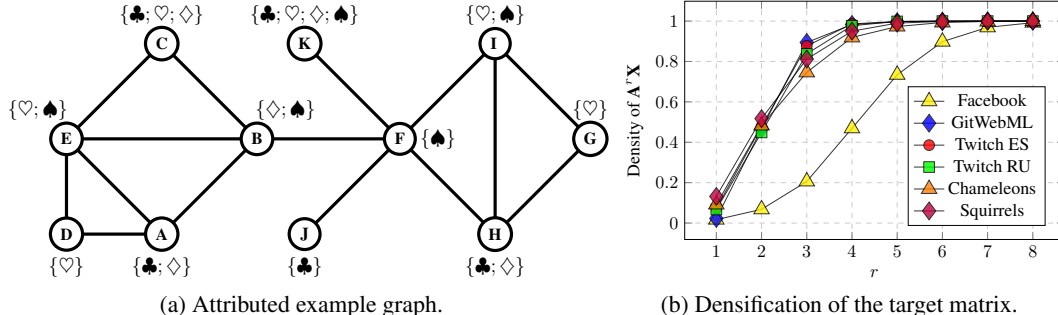

(a) Attributed example graph.  (b) Densification of the target matrix.

Figure 1: Phenomena affecting and inspiring the design of the multi-scale attributed network embedding procedure. In **Figure 1a** attributed nodes D and G have the same feature set and their nearest neighbours also exhibit equivalent sets of features, whereas features at higher order neighbourhoods differ. **Figure 1b** shows that as the order of neighbourhoods considered ($r$) increases, the product of the adjacency matrix power and the feature matrix becomes less sparse. This suggests that an implicit decomposition method would be computationally beneficial.

Our key contributions are:

1. to introduce the first Skip-gram style embedding algorithms that consider attribute distributions over local neighborhoods, both pooled (*AE*) and multi-scale (*MUSAE*), and their counterparts that attribute distinct features to each node (*AE-EGO* and *MUSAE-EGO*);

2. to theoretically prove that their embeddings approximately factorize PMI matrices based on the product of an adjacency matrix power and node-feature matrix;

3. to show that popular network embedding methods *DeepWalk* (Perozzi et al., 2014) and *Walklets* (Perozzi et al., 2017) are special cases of our *AE* and *MUSAE*;

4. we show empirically that *AE* and *MUSAE* embeddings enable strong performance at regression, classification, and link prediction tasks for real-world networks (e.g. Wikipedia and Facebook), are computationally scalable and enable transfer learning between networks.

We provide reference implementations of *AE* and *MUSAE*, together with the datasets used for evaluation at `https://github.com/iclr2020/MUSAE`.

## 2 RELATED WORK

Efficient unsupervised learning of node embeddings for large networks has seen unprecedented development in recent years. The current paradigm focuses on learning latent space representations of nodes such that those that share neighbors (Perozzi et al., 2014; Tang et al., 2015; Grover & Leskovec, 2016; Perozzi et al., 2017), structural roles (Ribeiro et al., 2017; Ahmed et al., 2018) or attributes are located close together in the embedding space. Our work falls under the last of these categories as our goal is to learn similar latent representations for nodes with similar sets of features in their neighborhoods, both on a pooled and multi-scale basis.

*Neighborhood preserving* node embedding procedures place nodes with common first, second and higher order neighbors within close proximity in the embedding space. Recent works in the neighborhood preserving node embedding literature were inspired by the *Skip-gram* model (Mikolov et al., 2013a;b), which generates word embeddings by implicitly factorizing a shifted pointwise mutual information (PMI) matrix (Levy & Goldberg, 2014) obtained from a text corpus. This procedure inspired *DeepWalk* (Perozzi et al., 2014), a method which generates truncated random walks over a graph to obtain a "corpus" from which the *Skip-gram* model generates neighborhood preserving node embeddings. In doing so, *DeepWalk* implicitly factorizes a PMI matrix, which can be shown, based on the underlying first-order Markov process, to correspond to the mean of a set of normalized adjacency matrix powers up to a given order (Qiu et al., 2018). Such *pooling* of matrices can be suboptimal since neighbors over increasing path lengths (or scales) are treated equally or according to fixed weightings (Mikolov et al., 2013a; Grover & Leskovec, 2016); whereas it has been found

that an optimal weighting may be task or dataset specific (Abu-El-Haija et al., 2018). In contrast, multi-scale node embedding methods such as *LINE* (Tang et al., 2015), *GraRep* (Cao et al., 2015) and *Walklets* (Perozzi et al., 2017) separately learn lower-dimensional node embedding components from each adjacency matrix power and concatenate them to form the full node representation. Such un-pooled representations, comprising distinct but less information at each scale, are found to give higher performance in a number of downstream settings, without increasing the overall number of free parameters (Perozzi et al., 2017).

*Attributed* node embedding procedures refine ideas from neighborhood based node embeddings to also incorporate node *attributes* (equivalently, features or labels) (Yang et al., 2015; Liao et al., 2018; Huang et al., 2017; Yang et al., 2018; Yang & Yang, 2018). Similarities between both a node's neighborhood structure and features contribute to determining pairwise proximity in the node embedding space. These models follow quite different strategies to obtain such representations. The most elemental procedure, *TADW* (Yang et al., 2015), decomposes a convex combination of normalized adjacency matrix powers into a matrix product that includes the feature matrix. Several other models, such as *SINE* (Zhang et al., 2018) and *ASNE* (Liao et al., 2018), implicitly factorize a matrix formed by concatenating the feature and adjacency matrices. Other approaches such as *TENE* (Yang & Yang, 2018), formulate the attributed node embedding task as a joint non-negative matrix factorization problem in which node representations obtained from sub-tasks are used to regularize one another. *AANE* (Huang et al., 2017) uses a similar network structure based regularization approach, in which a node feature similarity matrix is decomposed using the alternating direction method of multipliers. The method most similar to our own is *BANE* (Yang et al., 2018), in which the product of a normalized adjacency matrix power and a feature matrix is explicitly factorized to obtain attributed node embeddings. Many other methods exist, but do not consider the attributes of higher order neighborhoods (Yang et al., 2015; Liao et al., 2018; Huang et al., 2017; Zhang et al., 2018; Yang & Yang, 2018).

The relationship between our pooled (*AE*) and multi-scale (*MUSAE*) attributed node embedding methods mirrors that between graph convolutional neural networks (GCNNs) and multi-scale GC-NNs. Widely used graph convolutional layers, such as *GCN* (Kipf & Welling, 2017), *GraphSage* (Hamilton et al., 2017), *GAT* (Veličković et al., 2018), *APPNP* (Klicpera et al., 2019), *SGCONV* (Wu et al., 2019) and *ClusterGCN* (Chiang et al., 2019), create latent node representations that pool node attributes from arbitrary order neighborhoods, which are then inseparable and unrecoverable. In contrast, *MixHop* (Abu-El-Haija et al., 2019) learns latent features for each proximity.

## 3 ATTRIBUTED EMBEDDING MODELS

We now define algorithms to learn node embeddings using the attributes of nearby nodes, that allows both node and attribute embeddings to be learned jointly. The aim is to learn similar embeddings for nodes that occur in neighbourhoods of similar attributes; and similar embeddings for attributes that often occur in similar neighbourhoods of nodes. Let $\mathcal{G} = (\mathbb{V}, \mathbb{L})$ be an undirected graph of interest where $\mathbb{V}$ and $\mathbb{L}$ are the sets of vertices and edges (or links) respectively; and let $\mathbb{F}$ be the set of all possible node features (i.e. attributes). We define $\mathbb{F}_v \subseteq \mathbb{F}$ as the subset of features belonging to each node $v \in \mathbb{V}$. An embedding of nodes is a mapping $g : \mathbb{V} \to \mathbb{R}^d$ that assigns a $d$-dimensional representation $g(v)$ (or simply $g_v$) to each node $v$ and is fully described by a matrix $\boldsymbol{G} \in \mathbb{R}^{|\mathbb{V}| \times d}$. Similarly, an embedding of the features (to the same latent space) is a mapping $h : \mathbb{F} \to \mathbb{R}^d$ with embeddings denoted $h(f)$ (or simply $h_f$), and is fully described by a matrix $\boldsymbol{H} \in \mathbb{R}^{|\mathbb{F}| \times d}$.

### 3.1 ATTRIBUTED EMBEDDING

The *Attributed Embedding* (*AE*) procedure is described by Algorithm 1. We sample $n$ nodes $w_1$, from which to start attributed random walks on $\mathcal{G}$, with probability proportional to their degree (Line 2). From each starting node, a node sequence of length $l$ is sampled over $\mathcal{G}$ (Line 3), where sampling follows a first order random walk. For a given window size $t$, we iterate over each of the first $l-t$ nodes of the sequence termed *source* nodes $w_j$ (Line 4). For each source node, we consider the following $t$ nodes as *target* nodes (Line 5). For each target node $w_{j+r}$, we add the tuple $(w_j, f)$ to the corpus $\mathbb{D}$ for each target feature $f \in \mathbb{F}_{w_{j+r}}$ (Lines 6 and 7). We also consider features of the source node $f \in \mathbb{F}_{w_j}$, adding each $(w_{j+r}, f)$ tuple to $\mathbb{D}$ (Lines 9 and 10). Running Skip-gram on $\mathbb{D}$ with $b$ negative samples (Line 15) generates the $d$-dimensional node and feature embeddings.

**Data:** $\mathcal{G} = (\mathbb{V}, \mathbb{L})$ – Graph to be embedded.
   $\{\mathbb{F}_v\}_\mathbb{V}$ – Set of node feature sets.
   $n$ – Number of sequence samples.
   $l$ – Length of sequences.
   $t$ – Context size.
   $d$ – Embedding dimension.
   $b$ – Number of negative samples.
**Result:** Node embedding $g$ and feature embedding $h$.

1 **for** $i$ in $1:n$ **do**
2  Pick $w_1 \in \mathbb{V}$ according to $P(w_1) \sim deg(w_1)/\text{vol}(\mathcal{G})$.
3  $(w_1, w_2, \ldots, w_l) \leftarrow$ Sample Nodes$(\mathcal{G}, w_1, l)$
4  **for** $j$ in $1:l-t$ **do**
5   **for** $r$ in $1:t$ **do**
6    **for** $f$ in $\mathbb{F}_{w_{j+r}}$ **do**
7     Add tuple $(w_j, f)$ to multiset $\mathbb{D}$.
8    **end**
9    **for** $f$ in $\mathbb{F}_{w_j}$ **do**
10     Add tuple $(w_{j+r}, f)$ to multiset $\mathbb{D}$.
11    **end**
12   **end**
13  **end**
14 **end**
15 Run SGNS on $\mathbb{D}$ with $b$ negative samples and $d$ dimensions.
16 Output $g_v$, $\forall v \in \mathbb{V}$, and $h_f$, $\forall f \in \mathcal{F} = \cup_\mathbb{V}\mathbb{F}_v$.

**Algorithm 1:** AE sampling and training procedure

**Data:** $\mathcal{G} = (\mathbb{V}, \mathbb{L})$ – Graph to be embedded.
   $\{\mathbb{F}_v\}_\mathbb{V}$ – Set of node feature sets.
   $n$ – Number of sequence samples.
   $l$ – Length of sequences.
   $t$ – Context size.
   $d$ – Embedding dimension.
   $b$ – Number of negative samples.
**Result:** Node embeddings $g^r$ and feature embeddings $h^r$ for
   $r = 1, \ldots, t$.

1 **for** $i$ in $1:n$ **do**
2  Pick $w_1 \in \mathbb{V}$ according to $P(w_1) \sim deg(w_1)/\text{vol}(\mathcal{G})$.
3  $(w_1, w_2, \ldots, w_l) \leftarrow$ Sample Nodes$(\mathcal{G}, w_1, l)$
4  **for** $j$ in $1:l-t$ **do**
5   **for** $r$ in $1:t$ **do**
6    **for** $f$ in $\mathbb{F}_{w_{j+r}}$ **do**
7     Add the tuple $(w_j, f)$ to multiset $\mathbb{D}_{\underset{r}{\rightarrow}}$.
8    **end**
9    **for** $f$ in $\mathbb{F}_{w_j}$ **do**
10     Add the tuple $(w_{j+r}, f)$ to multiset $\mathbb{D}_{\underset{r}{\leftarrow}}$.
11    **end**
12   **end**
13  **end**
14 **end**
15 **for** $r$ in $1:t$ **do**
16  Create $\mathbb{D}_r$ by unification of $\mathbb{D}_{\underset{r}{\rightarrow}}$ and $\mathbb{D}_{\underset{r}{\leftarrow}}$.
17  Run SGNS on $\mathbb{D}_r$ with $b$ negative samples and $\frac{d}{t}$ dimensions.
18  Output $g_v^r$, $\forall v \in \mathbb{V}$, and $h_f^r$, $\forall f \in \mathbb{F} = \cup_\mathbb{V}\mathbb{F}_v$.
19 **end**

**Algorithm 2:** MUSAE sampling and training procedure

## 3.2 MULTI-SCALE ATTRIBUTED EMBEDDING

The *AE* method (Algorithm 1) pools feature sets of neighborhoods at different proximities. Inspired by the performance of (unattributed) multi-scale node embeddings, we adapt the *AE* algorithm to give *multi-scale attributed node embeddings* (*MUSAE*). The embedding component of a node $v \in \mathbb{V}$ for a specific proximity $r \in \{1, ..., t\}$ is given by a mapping $g^r : \mathbb{V} \rightarrow \mathbb{R}^{d/t}$ (assuming $t$ divides $d$). Similarly, the embedding component of feature $f \in \mathbb{F}$ at proximity $r$ is given by a mapping $h^r : \mathbb{F} \rightarrow \mathbb{R}^{d/t}$. Concatenating gives a $d$-dimensional embedding for each node and feature.

The *Multi-Scale Attributed Embedding* procedure is described by Algorithm 2. We again sample $n$ starting nodes $w_1$ with a probability proportional to node degree (Line 2) and, for each, sample a node sequence of length $l$ over $\mathcal{G}$ (Line 3) according to either a first or second order random walk. For a given window size $t$, we iterate over the first $l - t$ (source) nodes $w_j$ of the sequence (Line 4) and for each source node we iterate through the $t$ (target) nodes $w_{j+r}$ that follow (Line 5). We again consider each target node feature $f \in \mathbb{F}_{w_{j+r}}$, but now add tuples $(w_j, f)$ to a *sub-corpus* $\mathbb{D}_{\underset{r}{\rightarrow}}$ (Lines 6 and 7). We add tuples $(w_{j+r}, f)$ to another sub-corpus $\mathbb{D}_{\underset{r}{\leftarrow}}$ for each source node feature $f \in \mathbb{F}_{w_j}$ (Lines 9 and 10). Running Skip-gram on each sub-corpus $\mathbb{D}_r = \mathbb{D}_{\underset{r}{\rightarrow}} \cup \mathbb{D}_{\underset{r}{\leftarrow}}$ with $b$ negative samples (Line 16) output $t$ $(\frac{d}{t})$-dimensional node and feature embeddings that are concatenated.

## 4 ATTRIBUTED EMBEDDING AS IMPLICIT MATRIX FACTORIZATION

Levy & Goldberg (2014) showed that the loss function of Skip-gram with negative sampling (SGNS) is minimized if the embedding matrices factorize a matrix of pointwise mutual information (PMI) of word co-occurrence statistics. Specifically, for a word dictionary $\mathbb{V}$ with $|\mathbb{V}| = n$, SGNS (with $b$ negative samples) outputs two embedding matrices $\boldsymbol{W}, \boldsymbol{C} \in \mathbb{R}^{d \times n}$ such that $\forall w, c \in \mathbb{V}$:

$$\boldsymbol{w}_w^\top \boldsymbol{c}_c \approx \log\left(\frac{\#(w,c)|\mathbb{D}|}{\#(w)\#(c)}\right) - \log b \,,$$

where $\#(w, c)$, $\#(w)$, $\#(c)$ denote counts of word-context pair $(w, c)$, $w$ and $c$ over a corpus $\mathbb{D}$; and word embeddings $\boldsymbol{w}_w, \boldsymbol{c}_c \in \mathbb{R}^d$ are columns of $\boldsymbol{W}$ and $\boldsymbol{C}$ corresponding to $w$ and $c$ respectively. Considering $\frac{\#(w)}{|\mathbb{D}|}$, $\frac{\#(c)}{|\mathbb{D}|}$, $\frac{\#(w,c)}{|\mathbb{D}|}$ as empirical estimates of $p(w)$, $p(c)$ and $p(w, c)$ respectively shows:

$$\boldsymbol{W}^\top \boldsymbol{C} \approx \left[\text{PMI}(w, c) - \log b\right]_{w,c \in \mathbb{V}} \,,$$

i.e. an approximate low-rank factorization of a shifted PMI matrix (low rank since typically $d \ll n$).

Qiu et al. (2018) extended this result to node embedding models that apply SGNS to a "corpus" generated from random walks over the graph. In the case of *DeepWalk* where random walks are first-order Markov, the joint probability distributions over nodes at different stages of a random walk can be expressed in closed form. A closed form then follows for the factorized PMI matrix. We show that *AE* and *MUSAE* implicitly perform analogous matrix factorizations.

**Notation:** $\boldsymbol{A} \in \mathbb{R}^{n \times n}$ denotes the adjacency matrix and $\boldsymbol{D} \in \mathbb{R}^{n \times n}$ the diagonal degree matrix of a graph $\mathcal{G}$, i.e. $\boldsymbol{D}_{w,w} = \deg(w) = \sum_v \boldsymbol{A}_{w,v}$. We denote the *volume* of $\mathcal{G}$ by $c = \sum_{v,w} \boldsymbol{A}_{v,w}$. We define the binary attribute matrix $\boldsymbol{F} \in \{0,1\}^{|\mathbb{V}| \times |\mathbb{F}|}$ by $\boldsymbol{F}_{w,f} = \mathbf{1}_{f \in \mathbb{F}_w}, \ \forall w \in \mathbb{V}, f \in \mathbb{F}$. For ease of notation, we let $\boldsymbol{P} = \boldsymbol{D}^{-1} \boldsymbol{A}$ and $\boldsymbol{E} = diag(\mathbf{1}^\top \boldsymbol{D} \boldsymbol{F})$, where $diag$ indicates a diagonal matrix.

**Interpretation:** Assuming $\mathcal{G}$ is ergodic: $p(w) = \frac{deg(w)}{c}, w \in \mathbb{V}$ is the stationary distribution over nodes, i.e. $c^{-1} \boldsymbol{D} = diag(p(w))$; and $c^{-1} \boldsymbol{A}$ is the stationary joint distribution over consecutive nodes $p(w_j, w_{j+1})$. $\boldsymbol{F}_{w,f}$ can be considered a Bernoulli parameter describing the probability $p(f|w)$ of observing a feature $f$ at a node $w$ and so $c^{-1} \boldsymbol{D} \boldsymbol{F}$ describes the stationary joint distribution $p(f, w_j)$ over nodes and features. Accordingly, $\boldsymbol{P}$ is the matrix of conditional distributions $p(w_{j+1}|w_j)$; and $\boldsymbol{E}$ is a diagonal matrix proportional to the probability of observing each feature at the stationary distribution $p(f)$ (note that $p(f)$ need not sum to 1, whereas $p(w)$ necessarily must).

## 4.1 MULTI-SCALE CASE (MUSAE)

We know that the SGNS aspect of *MUSAE* (Algorithm 2, Line 17) is minimized when the learned embeddings $g_v^r, h_f^r$ satisfy $g_w^{r\top} h_f^r \approx \log \left( \frac{\#(w,f)_r |\mathbb{D}_r|}{\#(w)_r \#(f)_r} \right) - \log b \ \forall w \in \mathbb{V}, f \in \mathbb{F}$. Our aim is to express this factorization in terms of known properties of the graph $\mathcal{G}$ and its features.

**Lemma 1.** *The empirical statistics of node-feature pairs obtained from random walks give unbiased estimates of joint probabilities of observing feature $f \in \mathbb{F}$ $r$ steps (i) after; or (ii) before node $v \in \mathbb{V}$, as given by:*

$$\underset{l \to \infty}{plim} \frac{\#(w,f)_{\overrightarrow{r}}}{|\mathbb{D}_{\overrightarrow{r}}|} = c^{-1} (\boldsymbol{D} \boldsymbol{P}^r \boldsymbol{F})_{w,f} \qquad \underset{l \to \infty}{plim} \frac{\#(w,f)_{\overleftarrow{r}}}{|\mathbb{D}_{\overleftarrow{r}}|} = c^{-1} (\boldsymbol{F}^\top \boldsymbol{D} \boldsymbol{P}^r)_{f,w}$$

*Proof.* See Appendix. $\qquad \square$

**Lemma 2.** *Empirical statistics of node-feature pairs obtained from random walks give unbiased estimates of joint probabilities of observing feature $f \in \mathbb{F}$ $r$ steps either side of node $v \in \mathbb{V}$, given by:*

$$\underset{l \to \infty}{plim} \frac{\#(w,f)_r}{|\mathbb{D}_r|} = c^{-1} (\boldsymbol{D} \boldsymbol{P}^r \boldsymbol{F})_{w,f} \ ,$$

*Proof.* See Appendix. $\qquad \square$

Marginalizing gives unbiased estimates of stationary probability distributions of nodes and features:

$$\underset{l \to \infty}{plim} \frac{\#(w)}{|\mathbb{D}_r|} = \frac{deg(w)}{c} = c^{-1} \boldsymbol{D}_{w,w} \qquad \text{and} \qquad \underset{l \to \infty}{plim} \frac{\#(f)}{|\mathbb{D}_r|} = \sum_{w|f \in \mathbb{F}_w} \frac{deg(w)}{c} = c^{-1} \boldsymbol{E}_{f,f}$$

**Theorem 1.** *MUSAE embeddings approximately factorize the node-feature PMI matrix:*

$$\log \left( c \, \boldsymbol{P}^r \boldsymbol{F} \boldsymbol{E}^{-1} \right) - \log b, \quad for \, r = 1, \dots, t.$$

*Proof.*

$$\begin{aligned}
\frac{\#(w,f)_r |\mathbb{D}_r|}{\#(f)_r \#(w)_r} &= \left( \frac{\#(w,f)_r}{|\mathbb{D}_r|} \right) / \left( \frac{\#(f)_r}{|\mathbb{D}_r|} \frac{\#(w)_r}{|\mathbb{D}_r|} \right) \\
&\xrightarrow{p} \left( (c\boldsymbol{D}^{-1})(c^{-1} \boldsymbol{D} \boldsymbol{P}^r \boldsymbol{F})(c\boldsymbol{E}^{-1}) \right)_{w,f} \\
&= c(\boldsymbol{P}^r \boldsymbol{F} \boldsymbol{E}^{-1})_{w,f} \qquad\qquad\qquad \square
\end{aligned}$$

### 4.2 Pooled case (AE)

**Lemma 3.** *The empirical statistics of node-feature pairs learned by the AE algorithm give unbiased estimates of mean joint probabilities over different path lengths as follows:*

$$\underset{l\to\infty}{plim}\frac{\#(w,f)}{|\mathbb{D}|} = \frac{c}{t}\big(\boldsymbol{D}(\sum_{r=1}^{t}\boldsymbol{P}^r)\boldsymbol{F}\big)_{w,f} \tag{1}$$

*Proof.* By construction, $|\mathbb{D}|=\sum_r|\mathbb{D}_r|$, $\#(w,f)=\sum_r\#(w,f)_r$, $|\mathbb{D}_r|=|\mathbb{D}_s|\,\forall\,r,s\in\{1,\dots,t\}$ and so $|\mathbb{D}_s|=t^{-1}|\mathbb{D}|$. Combining with Lemma 2, the result follows. $\square$

**Theorem 2.** *AE embeddings approximately factorize the pooled node-feature matrix:*

$$\log\big(\tfrac{c}{t}(\sum_{r=1}^{t}\boldsymbol{P}^r)\boldsymbol{F}\boldsymbol{E}^{-1}\big) - \log b\,.$$

*Proof.* The proof is analogous to the proof of Theorem 1. $\square$

**Remark 1.** *DeepWalk is a corner case of AE with $\boldsymbol{F}=\boldsymbol{I}_{|\mathbb{V}|}$.*

That is, *DeepWalk* is equivalent to *AE* if each node has a single unique feature. Thus $\boldsymbol{E} = diag(\mathbf{1}^\top\boldsymbol{D}\boldsymbol{I})=\boldsymbol{D}$ and, by Theorem 2, *DeepWalk*'s embeddings factorize $\log\big(\frac{c}{t}(\sum_{r=1}^{t}\boldsymbol{P}^r)\mathbf{D}^{-1}\big) - \log b$, as previously noted by Qiu et al. (2018).

**Remark 2.** *Walklets is a corner case of MUSAE with $\boldsymbol{F}=\boldsymbol{I}_{|\mathbb{V}|}$.*

Thus, for $r=1,\dots,t$, the embeddings of *Walklets* factorise $\log\big(c\,\mathbf{P}^r\mathbf{D}^{-1}\big) - \log b$.

**Remark 3.** *Appending an identity matrix $\boldsymbol{I}$ to the feature matrices $\boldsymbol{F}$ of AE and MUSAE (denoted $[\boldsymbol{F};\boldsymbol{I}]$) adds a unique feature to each node. The resulting algorithms, named AE-EGO and MUSAE-EGO, learn embeddings that, respectively, approximately factorize the node-feature PMI matrices:*

$$log\big(c\,\boldsymbol{P}^r\,[\boldsymbol{F};\boldsymbol{I}]\,\boldsymbol{E}^{-1}\big) - \log b,\ \ \forall r\in\{1,...,t\}; \qquad and \qquad \log\big(\tfrac{c}{t}(\sum_{r=1}^{t}\boldsymbol{P}^r)\,[\boldsymbol{F};\boldsymbol{I}]\,\boldsymbol{E}^{-1}\big) - \log b\,.$$

### 4.3 Complexity analysis

Under the assumption of a constant number of features per source node and first-order attributed random walk sampling, the corpus generation has a runtime complexity of $\mathcal{O}(n\,l\,t\,x/y)$, where $x = \sum_{v\in\mathbb{V}}|\mathbb{F}_v|$ the total number of features across all nodes (including repetition) and $y = |\mathbb{V}|$ the number of nodes. Using negative sampling, the optimization runtime of a single asynchronous gradient descent epoch on *AE* and the joint optimization runtime of *MUSAE* embeddings is described by $\mathcal{O}(b\,d\,n\,l\,t\,x/y)$. If one does $p$ truncated walks from each source node, the corpus generation complexity is $\mathcal{O}(p\,y\,l\,t\,x)$ and the model optimization runtime is $\mathcal{O}(b\,d\,p\,y\,l\,t\,x)$. Our later runtime experiments in Section 5 will underpin optimization runtime complexity discussed above.

Corpus generation has a memory complexity of $\mathcal{O}(n\,l\,t\,x/y)$ while the same when generating $p$ truncated walks per node has a memory complexity of $\mathcal{O}(p\,y\,l\,t\,x)$. Storing the parameters of an *AE* embedding has a memory complexity of $\mathcal{O}(y\,d)$ and *MUSAE* embeddings also use $\mathcal{O}(y\,d)$ memory.

## 5 Experimental Evaluation

In order to evaluate the quality of created representations we test the embeddings on supervised downstream tasks such as node classification, transfer learning across networks, regression, and link prediction. Finally, we investigate how changes in the input size affect the runtime. For doing so we utilize social networks and web graphs that we collected from Facebook, Github, Twitch and Wikipedia. The data sources, collection procedures and the datasets themselves are described with great detail in Appendix B. In addition we tested our methods on citation networks widely used for model evaluation (Shchur et al., 2018). Across all experiments we use the same hyperparameter settings of our own model, competing unsupervised methods and graph neural networks – these are respectively listed in Appendices C, E and F.

## 5.1 NODE CLASSIFICATION

We evaluate the node classification performance in two separate scenarios. In the first we do $k$-shot learning by using the attributed embedding vectors with logistic regression to predict labels on the Facebook, Github and Twitch Portugal graphs. In the second we test the predictive performance under a fixed size train-test split to compare against various embedding methods and competitive neural network architectures.

### 5.1.1 k-SHOT LEARNING

In this experiment we take $k$ randomly selected samples per class, and use the attributed node embeddings to train a logistic regression model with $l_2$ regularization and predict the labels on the remaining vertices. We repeated the above procedure with seeded splits 100 times to obtain robust comparable results (Shchur et al., 2018). From these we calculated the average of micro averaged $F_1$ scores to compare our own methods with other unsupervised node embedding procedures. We varied $k$ in order to show the efficacy of the methods – what are the gains when the training set size is increased. These results are plotted in Figure 2 for Facebook, Github and Twitch Portugal networks.

Based on these plots it is evident that *MUSAE* and *AE* embeddings have little gains in terms of micro $F_1$ score when additional data points are added to the training set when $k$ is larger than 12. This implies that our method is data efficient. Moreover, *MUSAE-EGO* and *AE-EGO* have a slight performance advantage, which means that including the nodes in the attributed random walks helps when a small amount of labeled data is available in the downstream task.

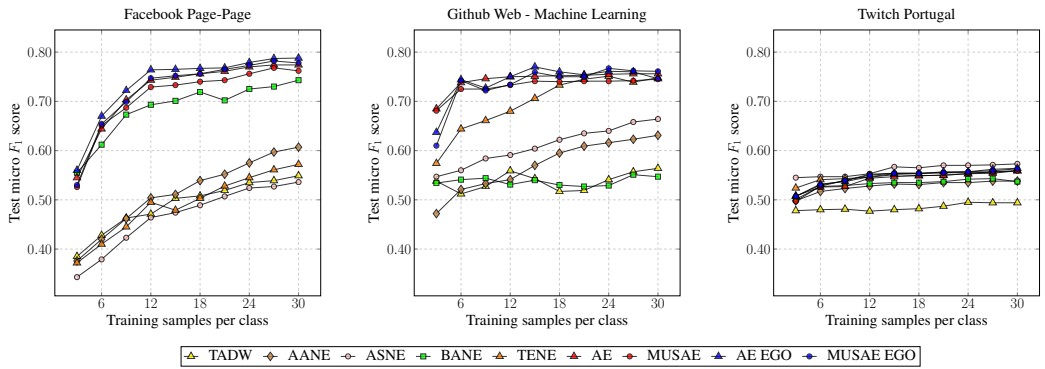

Figure 2: Node classification $k$-shot learning performance as a function of training samples per class evaluated by average micro $F_1$ scores calculated from a 100 seeded train-test splits.

### 5.1.2 FIXED RATIO TRAIN-TEST SPLITS

In this series of experiments we created a 100 seeded train test splits of nodes (80% train - 20% test) and calculated weighted, micro and macro averaged $F_1$ scores on the test set to compare our methods to various embedding and graph neural network methods. Across procedures the same random seeds were used to obtain the train-test split this way the performances are directly comparable. We attached these results on the Facebook, Github and Twitch Portugal graphs as Table 6 of Appendix G. In each column red denotes the best performing unsupervised embedding model and blue corresponds to the strongest supervised neural model. We also attached additional supporting results using the same experimental setting with the unsupervised methods on the Cora, Citeseer, and Pubmed graphs as Table 5 of Appendix G.

In terms of micro $F_1$ score our strongest method outperforms on the Facebook and GitHub networks the best unsupervised method by 1.01% and 0.47% respectively. On the Twitch Portugal network the relative micro $F_1$ advantage of *ASNE* over our best method is 1.02%. Supervised node embedding methods outperform our and other unsupervised methods on every dataset for most metrics. In terms of micro $F_1$ this relative advantage over our best performing model variant is the largest with 4.67% on the Facebook network, and only 0.11% on Twitch Portugal.

One can make four general observations based on our results (i) multi-scale representations can help with the classification tasks compared to pooled ones; (ii) the addition of the nodes in the ego augmented models to the feature sets does not help the performance when a large amount of labeled

training data is available; (iii) based on the standard errors supervised neural models do not necessarily have a significant advantage over unsupervised methods (see the results on the Github and Twitch datasets); (iv) attributed node embedding methods that only consider first-order neighbourhoods have a poor performance.

## 5.2 Transfer Learning on twitch social networks

Neighbourhood based methods such as *DeepWalk* (Perozzi et al., 2014) are transductive and the function used to create the embedding cannot map nodes that are not connected to the original graph to the latent space. However, vanilla *MUSAE* and *AE* are inductive and can easily map nodes to the embedding space if the attributes across the source and target graph are shared. This also means that supervised models trained on the embedding of a source graph are transferable. Importantly those attributed embedding methods such as *AANE* or *ASNE* that explicitly use the graph are unable to do this transfer.

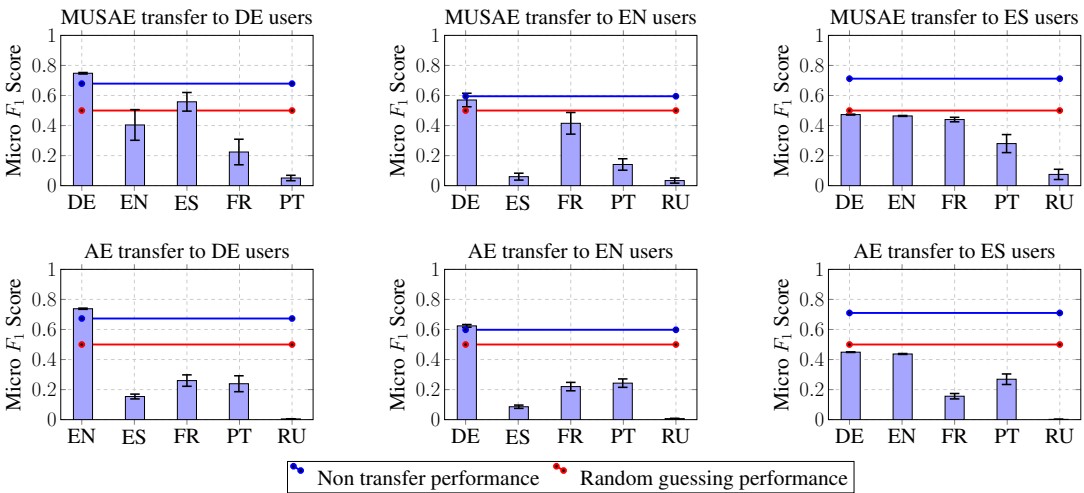

Figure 3: Mean micro $F_1$ scores and standard errors calculated from 10 transfer learning runs with *MUSAE* and *AE* on the Twitch graphs using Germany, England and Spain as target for the transfer. The blue reference line denotes the test performance on the target dataset in a non transfer learning scenario (standard hyperparameter settings and split ratio). The red reference line denotes the performance of random guesses.

Using the disjoint Twitch country level social networks (inter country edges are not present) we did a transfer learning experiment. First, we learn an embedding function given the social network from a country with the standard parameter settings. Second, we train regularized logistic regression on the embedding to predict whether the Twitch user streams explicit content. Third, using the embedding function we map the target graph to the embedding space. Fourth, we use the logistic model to predict the node labels on the target graph. We evaluate the performance by the micro $F_1$ score based on 10 experimental repetitions. These averages with standard error bars are plotted for the Twitch Germany, England and Spain datasets as target graphs on Figure 3. We added additional results with France, Portugal and Russia being the target country in Appendix H as Table 5.

These results support that *MUSAE* and *AE* create features that are transferable across graphs that share vertex features. For example, based on a comparison to non transfer-learning results we find that the transfer between the German and English user graphs is effective in terms of micro $F_1$ score. Transfer from English users to German ones considerably improves performance, and the other way around there is a little gain. We also see that the upstream and downstream models that we trained on graphs with more vertices transfer well while transfer to the small ones is generally poor – most of the times worse than random guessing. There is no clear evidence that either *MUSAE* or *AE* gives better results on this specific problem.

## 5.3 Regression on Wikipedia Graphs

We created embeddings of the Wikipedia webgraphs with all of our methods and the unsupervised baselines. Using a 80% train - 20% test split we predict the log of average traffic for each page using

an elastic net model. The hyperparameters of the downstream model are available in Appendix D. In Table 7 of Appendix I we report average test $R^2$ and standard error of the predictive performance over 100 seeded train-test splits. Our key observation are: (i) that *MUSAE* outperforms all benchmark neighbourhood preserving and attributed node embedding methods, with the strongest *MUSAE* variant outperforming the best baseline between 2.05% and 10.03% (test $R^2$); (ii) that *MUSAE* significantly outperforms *AE* by between 2.49% and 21.64% (test $R^2$); and (iii) the benefit of using the vertices as features (ego augmented model) can improve the performance of embeddings, but appears to be dataset specific phenomenon.

### 5.4 LINK PREDICTION ON WEB GRAPHS AND SOCIAL NETWORKS

The final series of experiments dedicated to the representation quality is about link prediction. We carried out an attenuated graph embedding trial to predict the removed edges from the graph. First, we randomly removed 50% of edges while the connectivity of the graph was not changed. Second, an embedding is created from the attenuated graph. Third, we calculate features for the removed edges and the same number of randomly selected pairs of nodes (negative candidates) with binary operators to create $d$-dimensional edge features. We use the binary operators applied by Grover & Leskovec (2016). Specifically, we calculated the average, element-wise product, element-wise $l_1$ norm and the element-wise $l_2$ norm of vectors. Finally, we created a 100 seeded 80% train - 20% test splits and used logistic regression to predict whether an edge exists.

We compared to attributed and neighbourhood based embedding methods and average AUC scores are presented in Tables 8 and 9 of Appendix J. Our results show that *Walklets* (Perozzi et al., 2017) the multi-scale neighbourhood based embedding method materially outperforms every other method on most of the datasets and attributed embedding methods generally do poorly in terms of AUC compared to neighbourhood based ones.

### 5.5 SCALABILITY

In order to show the efficacy of our algorithms we run a series of experiments on synthetic graphs where we are able to manipulate the input size. Specifically, we look at the effect of changing the number of vertices and features per vertex. Our detailed experimental setup was as follows. Each point in Figure 4 is the mean runtime obtained from 100 experimental runs on Erdos-Renyi graphs. The base graph that we manipulated had $2^{11}$ nodes, $2^3$ edges and the same number of unique features per node uniformly selected from a feature set of $2^{11}$. Our experimental settings were the same as the ones described in Appendix C except for the number of epochs. We only did a single training epoch with asynchronous gradient descent on each graph. We tested the runtime with 1, 2 and 4 cores and included a dashed line as the linear runtime reference in each subfigure.

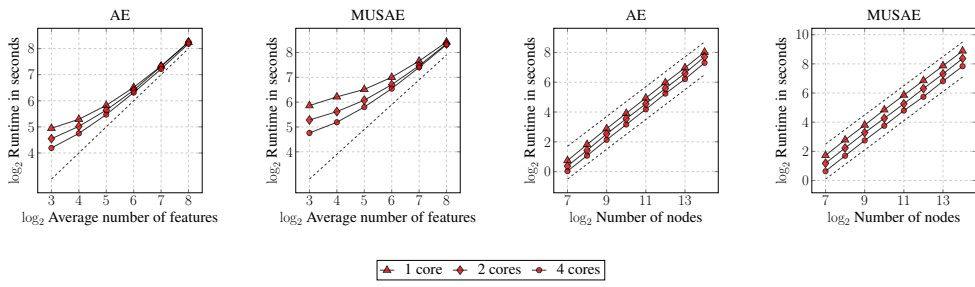

Figure 4: Optimization time as a function of average feature count / number of vertices.

We observe that doubling the average number of features per vertex doubles the runtime of *AE* and *MUSAE*. Moreover, the number of cores used during the optimization does not decrease the runtime when the number of unique features per vertex compared to the cardinality of the feature set is large. When we look at the change in the vertex set size we also see a linear behaviour. Doubling the input size simply results in a doubled optimization runtime. In addition, if one interpolates linearly from these results it comes that a network with 1 million nodes, 8 edges per node, 8 unique features per node can be embedded with *MUSAE* on commodity hardware in less than 5 hours. This

interpolation assumes that the standard parameter settings proposed in Appendix C and 4 cores were used for optimization.

## 6 DISCUSSION AND CONCLUSION

We investigated attributed node embedding and proposes efficient pooled (*AE*) and multi-scale (*MUSAE*) attributed node embedding algorithms with linear runtime. We proved that these algorithms implicitly factorize probability matrices of features appearing in the neighbourhood of nodes. Two widely used neighbourhood preserving node embedding methods Perozzi et al. (2014; 2017) are in fact simplified cases of our models. On several datasets (Wikipedia, Facebook, Github, and citation networks) we found that representations learned by our methods, in particular *MUSAE*, outperform neighbourhood based node embedding methods (Perozzi et al. (2014); Grover & Leskovec (2016)), multi-scale algorithms (Tang et al. (2015); Perozzi et al. (2017)) and recently proposed attributed node embedding procedures (Yang et al. (2015); Liao et al. (2018); Huang et al. (2017); Yang et al. (2018); Yang & Yang (2018)).

Our proposed embedding models are differentiated from other methods in that they encode feature information from higher order neighborhoods. The most similar previous model *BANE* (Yang et al., 2018) encodes node attributes from higher order neighbourhoods but has non-linear runtime complexity and the product of adjacency matrix power and feature matrix is decomposed explicitly.

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

## A  PROOFS

**Lemma 1.** *The empirical statistics of node-feature pairs obtained from random walks give unbiased estimates of joint probabilities of observing feature $f \in \mathbb{F}$ r steps (i) after; or (ii) before node $v \in \mathbb{V}$, as given by:*

$$\underset{l\to\infty}{plim}\frac{\#(w,f)_{\overrightarrow{r}}}{|\mathbb{D}_{\overrightarrow{r}}|} = c^{-1}(\boldsymbol{D}\boldsymbol{P}^r\boldsymbol{F})_{w,f} \qquad \underset{l\to\infty}{plim}\frac{\#(w,f)_{\overleftarrow{r}}}{|\mathbb{D}_{\overleftarrow{r}}|} = c^{-1}(\boldsymbol{F}^\top\boldsymbol{D}\boldsymbol{P}^r)_{f,w}$$

*Proof.* The proof is analogous to that given for Theorem 2.1 in Qiu et al. (2018). We show that the computed statistics correspond to sequences of random variables with finite expectation, bounded variance and covariances that tend to zero as the separation between variables within the sequence tends to infinity. The Weak Law of Large Numbers (S.N.Bernstein) then guarantees that the sample mean converges to the expectation of the random variable. We first consider the special case $n = 1$, i.e. we have a single sequence $w_1, ..., w_l$ generated by a random walk (see Algorithm 1). For a particular node-feature pair $(w, f)$, we let $Y_i$, $i \in \{1, ..., l - t\}$, be the indicator function for the event $w_i = w$ and $f \in \mathbb{F}_{i+r}$. Thus, we have:

$$\frac{\#(w,f)_{\overrightarrow{r}}}{|\mathbb{D}_{\overrightarrow{r}}|} = \frac{1}{l-t}\sum_{i=1}^{l-t}Y_i, \tag{2}$$

the sample average of the $Y_i$s. We also have:

$$\mathbb{E}[Y_i] = \frac{deg(w)}{c}(\boldsymbol{P}^r\boldsymbol{F})_{w,f} = \frac{1}{c}(\boldsymbol{D}\boldsymbol{P}^r\boldsymbol{F})_{w,f}$$

$$\mathbb{E}[Y_iY_j] = \text{Prob}[w_i = w, f \in \mathbb{F}_{i+r}, w_j = w, f \in \mathbb{F}_{j+r}]$$

$$= \underbrace{\frac{deg(w)}{c}}_{p(w_i=w)}\underbrace{\underbrace{\boldsymbol{P}^r_{:w}}_{p(w_{i+r}|w_i=w)}\underbrace{diag(\boldsymbol{F}_{:f})}_{p(f\in\mathbb{F}_{i+r}|w_{i+r})}\underbrace{\boldsymbol{P}^{j-(i+r)}_{:w}}_{p(w_j=w|w_{i+r})}}_{p(w_j=w,f\in\mathbb{F}_{i+r}|w_i=w)}\underbrace{\boldsymbol{P}^r_{w:}\boldsymbol{F}_{:f}}_{p(f\in\mathbb{F}_{j+r}|w_j=w)}$$

for $j > i + r$. This allows us to compute the covariance:

$$\text{Cov}(Y_i, Y_j) = \mathbb{E}[Y_iY_j] - \mathbb{E}[Y_i]\,\mathbb{E}[Y_j]$$
$$= \frac{deg(w)}{c}\boldsymbol{P}^r_{w:}diag(\boldsymbol{F}_{:f})\underbrace{\left(\boldsymbol{P}^{j-(i+r)}_{:w} - \frac{deg(w)}{c}\underline{1}\right)}_{\text{tends to 0 as } j-i\to\infty}\boldsymbol{P}^r_{w:}\boldsymbol{F}_{:f}, \tag{3}$$

where $\underline{1}$ is a vector of ones. The difference term (indicated) tends to zero as $j - i \to \infty$ since then $p(w_j = w|w_{i+r})$ tends to the stationary distribution $p(w) = \frac{deg(w)}{c}$, regardless of $w_{i+r}$. Thus, applying the Weak Law of Large Numbers, the sample average converges in probability to the expected value, i.e.:

$$\frac{\#(w,f)_{\overrightarrow{r}}}{|\mathbb{D}_{\overrightarrow{r}}|} = \frac{1}{l-t}\sum_{i=1}^{l-t}Y_i \xrightarrow{p} \frac{1}{l-t}\sum_{i=1}^{l-t}\mathbb{E}[Y_i] = \frac{1}{c}(\boldsymbol{D}\boldsymbol{P}^r\boldsymbol{F})_{w,f}$$

A similar argument applies to $\frac{\#(w,f)_{\overleftarrow{r}}}{|\mathbb{D}_{\overleftarrow{r}}|}$, with expectation term $\frac{1}{c}(\boldsymbol{F}^\top\boldsymbol{D}\boldsymbol{P}^r)_{f,w}$. In both cases, the argument readily extends to the general setting where $n > 1$ with suitably defined indicator functions for each of the $n$ random walks (see Qiu et al. (2018)). □

**Lemma 2.** *Empirical statistics of node-feature pairs obtained from random walks give unbiased estimates of joint probabilities of observing feature $f \in \mathbb{F}$ $r$ steps either side of node $v \in \mathbb{V}$, given by:*

$$\underset{l \to \infty}{plim} \frac{\#(w,f)_r}{|\mathbb{D}_r|} = c^{-1}(\boldsymbol{D}\boldsymbol{P}^r\boldsymbol{F})_{w,f} \, ,$$

*Proof.*

$$
\begin{aligned}
\frac{\#(w,f)_r}{|\mathbb{D}_r|} &= \frac{\#(w,f)_{\overrightarrow{r}}}{|\mathbb{D}_r|} + \frac{\#(w,f)_{\overleftarrow{r}}}{|\mathbb{D}_r|} \\
&= \tfrac{1}{2}\Big(\frac{\#(w,f)_{\overrightarrow{r}}}{|\mathbb{D}_{\overrightarrow{r}}|} + \frac{\#(w,f)_{\overleftarrow{r}}}{|\mathbb{D}_{\overleftarrow{r}}|}\Big) \\
&\xrightarrow{p} \tfrac{1}{2}\Big(\tfrac{1}{c}(\boldsymbol{D}\boldsymbol{P}^r\boldsymbol{F})_{w,f} + \tfrac{1}{c}(\boldsymbol{F}^\top\boldsymbol{D}\boldsymbol{P}^r)_{f,w}\Big) \\
&= \tfrac{1}{2c}\big(\boldsymbol{D}\boldsymbol{P}^r\boldsymbol{F} + \boldsymbol{P}^{r\top}\boldsymbol{D}\boldsymbol{F}\big)_{w,f} \\
&= \tfrac{1}{2c}\big((\boldsymbol{D}\boldsymbol{P}^r + (\boldsymbol{A}^\top\boldsymbol{D}^{-1})^r\boldsymbol{D})\boldsymbol{F}\big)_{w,f} \\
&= \tfrac{1}{2c}\big((\boldsymbol{D}\boldsymbol{P}^r + \boldsymbol{D}(\boldsymbol{D}^{-1}\boldsymbol{A}^\top)^r)\boldsymbol{F}\big)_{w,f} \\
&= \tfrac{1}{c}(\boldsymbol{D}\boldsymbol{P}^r\boldsymbol{F})_{w,f} \, .
\end{aligned}
$$

The final step follows by symmetry of $\boldsymbol{A}$, indicating how the Lemma can be extended to directed graphs. $\qquad\square$

## B    DATASETS AND DESCRIPTIVE STATISTICS

Our method was evaluated on a variety of social networks and web page-page graphs that we collected from openly available API services. In Table 1 we described the graphs with widely used statistics with respect to size, diameter, and level of clustering. We also included the average number of features per vertex and unique feature count in the last columns. These datasets are available with the source code of *MUSAE* and *AE* at `https://github.com/iclr2020/MUSAE`.

Table 1: Descriptive statistics of the networks used in our experimental evaluation.

| Dataset | Nodes | Edges | Diameter | Clustering Coefficient | Density | Average Feature | Unique Features |
|---|---|---|---|---|---|---|---|
| Facebook Page-Page | 22,470 | 171,002 | 15 | 0.232 | 0.001 | 14.000 | 4,714 |
| GitHub Web-ML | 37,700 | 289,003 | 7 | 0.013 | 0.001 | 18.312 | 4,005 |
| Wikipedia Chameleon | 2,277 | 31,421 | 11 | 0.314 | 0.012 | 21.547 | 3,132 |
| Wikipedia Crocodile | 11,631 | 170,918 | 11 | 0.026 | 0.003 | 75.161 | 13,183 |
| Wikipedia Squirrel | 5,201 | 198,493 | 10 | 0.348 | 0.015 | 26.474 | 3,148 |
| Twitch DE | 9,498 | 153,138 | 7 | 0.047 | 0.003 | 20.397 | 2,545 |
| Twitch EN | 7,126 | 35,324 | 10 | 0.042 | 0.002 | 20.799 | 2,545 |
| Twitch ES | 4,648 | 59,382 | 9 | 0.084 | 0.006 | 19.391 | 2,545 |
| Twitch FR | 6,549 | 112,666 | 7 | 0.054 | 0.005 | 19.758 | 2,545 |
| Twitch PT | 1,912 | 31,299 | 7 | 0.131 | 0.017 | 19.944 | 2,545 |
| Twitch RU | 4,385 | 37,304 | 9 | 0.049 | 0.004 | 20.635 | 2,545 |

### B.1    FACEBOOK PAGE-PAGE DATASET

This webgraph is a page-page graph of verified Facebook sites. Nodes represent official Facebook pages while the links are mutual likes between sites. Node features are extracted from the site descriptions that the page owners created to summarize the purpose of the site. This graph was collected through the Facebook Graph API in November 2017 and restricted to pages from 4 categories which are defined by Facebook. These categories are: *politicians*, *governmental organizations*, *television shows* and *companies*. As one can see in Table 1 it is a highly clustered graph with a large diameter. The task related to this dataset is multi-class node classification for the 4 site categories.

### B.2 GitHub web and machine learning developers dataset

The largest graph used for evaluation is a social network of GitHub developers which we collected from the public API in June 2019. Nodes are developers who have starred at least 10 repositories and edges are mutual follower relationships between them. The vertex features are extracted based on the location, repositories starred, employer and e-mail address. The task related to the graph is binary node classification – one has to predict whether the GitHub user is a web or a machine learning developer. This target feature was derived from the job title of each user. As the descriptive statistics show in Table 1 this is the largest graph that we use for evaluation with the highest sparsity.

### B.3 Wikipedia datasets

The datasets that we use to perform node level regression are Wikipedia page-page networks collected on three specific topics: chameleons, crocodiles and squirrels. In these networks nodes are articles from the English Wikipedia collected in December 2018, edges are mutual links that exist between pairs of sites. Node features describe the presence of nouns appearing in the articles. For each node we also have the average monthly traffic between October 2017 and November 2018. In the regression tasks used for embedding evaluation the logarithm of average traffic is the target variable. Table 1 shows that these networks are heterogeneous in terms of size, density, and clustering.

### B.4 Twitch datasets

These datasets used for node classification and transfer learning are Twitch user-user networks of gamers who stream in a certain language. Nodes are the users themselves and the links are mutual friendships between them. Vertex features are extracted based on the games played and liked, location and streaming habits. Datasets share the same set of node features, this makes *transfer learning* across networks possible. These social networks were collected in May 2018. The supervised task related to these networks is binary node classification – one has to predict whether a streamer uses explicit language.

## C Standard hyperparameter settings of our embedding models

In *MUSAE* and *AE* models we have a set of parameters that we use for model evaluation. Our parameter settings listed in Table 2 are quite similar to the widely used general settings of random walk sampled implicit factorization machines (Perozzi et al., 2014; Grover & Leskovec, 2016; Ribeiro et al., 2017; Perozzi et al., 2017). Each of our models is augmented with a *Doc2Vec* (Mikolov et al., 2013a;b) embedding of node features – this is done such way that the overall dimension is still 128.

Table 2: Standard hyperparameter settings of the AE and MUSAE embeddings.

| Parameter | Value | Notation |
|---|---|---|
| Dimensions | 128 | $d$ |
| Walk length | 80 | $l$ |
| Number of walks per node | 10 | $p$ |
| Number of epochs | 5 | $k$ |
| Window size | 3 | $t$ |
| Initial learning rate | 0.05 | $\alpha_{max}$ |
| Final learning rate | 0.025 | $\alpha_{min}$ |
| Negative samples | 5 | $b$ |

## D Hyperparameter settings of the downstream models

The downstream tasks uses logistic and elastic net regression from *Scikit-learn* (Pedregosa et al., 2011) for node level classification, regression and link prediction. For the evaluation of every embedding model we use the standard settings of the library except for the regularization and norm mixing parameters. These are described in Table 3.

## E Hyperparameter settings of competing unsupervised embedding methods

Our purpose was a fair evaluation compared to other node embedding procedures. Because of this each we tried to use hyperparameter settings that give similar expressive power to the competing

Table 3: Standard hyperparameter settings of the downstream logistic and elastic net regression models that use the embeddings for classification, link prediction and regression.

| Parameter | Value | Notation |
|---|---|---|
| Regularization coefficient | 0.01 | $\lambda$ |
| Norm mixing parameter | 0.5 | $\gamma$ |

methods with respect to target matrix approximation (Perozzi et al., 2014; Grover & Leskovec, 2016; Perozzi et al., 2017) and number of dimensions.

- *DeepWalk* (Perozzi et al., 2014): We used the hyperparameter settings described in Table 2. While the original *DeepWalk* model uses hierarchical softmax to speed up calculations we used a negative sampling based implementation. This way *DeepWalk* can be seen as a special case of *Node2Vec* (Grover & Leskovec, 2016) when the second-order random walks are equivalent to the firs-order walks.

- *LINE$_2$* (Tang et al., 2015): We created 64 dimensional embeddings based on first and second order proximity and concatenated these together for the downstream tasks. Other hyperparameters are taken from the original work.

- *Node2Vec* (Grover & Leskovec, 2016): Except for the *in-out* and *return* parameters that control the second-order random walk behavior we used the hyperparameter settings described in Table 2. These behavior control parameters were tuned with grid search from the $\{4, 2, 1, 0.5, 0.25\}$ set using a train-validation split of $80\% - 20\%$ *within the training set* itself.

- *Walklets* (Perozzi et al., 2017): We used the hyperparameters described in Table 2 except for window size. We set a window size of 4 with individual embedding sizes of 32. This way the overall number of dimensions of the representation remained the same.

- The attributed node embedding methods *AANE*, *ASNE*, *BANE*, *TADW*, *TENE* all use the hyperparameters described in the respective papers except for the dimension. We parametrized these methods such way that each of the final embeddings used in the downstream tasks is 128 dimensional.

## F HYPERPARAMETER SETTINGS OF COMPETING GRAPH NEURAL NETWORKS

Each model was optimized with the Adam optimizer (Kingma & Ba, 2015) with the standard moving average parameters and the model implementations are sparsity aware modifications based on PyTorch Geometric (Fey & Lenssen, 2019). We needed these modifications in order to accommodate the large number of vertex features – see the last column in Table 1. Except for the *GAT* model (Veličković et al., 2018) we used ReLU intermediate activation functions (Nair & Hinton, 2010) with a softmax unit in the final layer for classification. The hyperparameters used for the training and regularization of the neural models are listed in Table 4.

Table 4: Hyperparameter settings used for training the graph neural network baselines.

| Parameter | Value |
|---|---|
| Epochs | 200 |
| Learning rate | 0.01 |
| Dropout | 0.5 |
| $l_2$ Weight regularization | 0.001 |
| Depth | 2 |
| Filters per layer | 32 |

Except for the *APPNP* model each baseline uses information up to 2-hop neighbourhoods. The model specific settings when we needed to deviate from the basic settings which are listed in Table 4 were as follows:

- *Classical GCN* (Kipf & Welling, 2017): We used the standard parameter settings described in this section.

- *GraphSAGE* (Hamilton et al., 2017): We utilized a graph convolutional aggregator on the sampled neighbourhoods, samples of 40 nodes per source, and standard settings.

- *GAT* (Veličković et al., 2018): The negative slope parameter of the leaky ReLU function was 0.2, we applied a single attention head, and used the standard hyperparameter settings.

- *MixHop* (Abu-El-Haija et al., 2019): We took advantage of the $0^{th}$, $1^{st}$ and $2^{nd}$ powers of the normalized adjacency matrix with 32 dimensional convolutional filters for creating the first hidden representations. This was fed to a feed-forward layer to classify the nodes.

- *ClusterGCN* (Chiang et al., 2019): Just as Chiang et al. (2019) did, we used the *METIS* procedure (Karypis & Kumar, 1998). We clustered the graphs into disjoint clusters, and the number of clusters was the same as the number of node classes (e.g. in case of the Facebook page-page network we created 4 clusters). For training we used the earlier described setup.

- *APPNP* (Klicpera et al., 2019): The top level feed-forward layer had 32 hidden neurons, the teleport probability was set as 0.2 and we used 20 steps for approximate personalized pagerank calculation.

- *SGCONV* (Wu et al., 2019): We used the $2^{nd}$ power of the normalized adjacency matrix for training the classifier.

## G CLASSIFICATION PERFORMANCE

Table 5: Node classification test performance evaluated by weighted, micro and macro $F_1$ scores calculated from 10 seeded train-test splits. We included standard errors of the scores and used 80% of nodes for training / 20% of nodes for testing. Red numbers denote the best performing node embedding method.

| | Datasets | | | | | | | | |
| | Cora | | | Citeseer | | | Pubmed | | |
| | Weighted | Micro | Macro | Weighted | Micro | Macro | Weighted | Micro | Macro |
|---|---|---|---|---|---|---|---|---|---|
| **DeepWalk** | 0.832 ±0.003 | 0.833 ±0.004 | 0.823 ±0.004 | 0.597 ±0.007 | 0.603 ±0.007 | 0.560 ±0.006 | 0.801 ±0.001 | 0.802 ±0.001 | 0.789 ±0.002 |
| **LINE$_2$** | 0.775 ±0.004 | 0.777 ±0.004 | 0.768 ±0.005 | 0.529 ±0.006 | 0.542 ±0.006 | 0.486 ±0.005 | 0.798 ±0.001 | 0.799 ±0.001 | 0.785 ±0.001 |
| **Node2Vec** | 0.840 ±0.003 | 0.840 ±0.003 | 0.826 ±0.003 | 0.616 ±0.005 | 0.622 ±0.005 | 0.581 ±0.005 | 0.809 ±0.002 | 0.810 ±0.002 | 0.797 ±0.002 |
| **Walklets** | 0.843 ±0.003 | 0.843 ±0.003 | 0.827 ±0.003 | 0.624 ±0.005 | 0.630 ±0.006 | 0.590 ±0.005 | 0.815 ±0.001 | 0.815 ±0.001 | 0.804 ±0.002 |
| **TADW** | 0.819 ±0.004 | 0.819 ±0.004 | 0.804 ±0.005 | 0.725 ±0.004 | 0.734 ±0.004 | 0.685 ±0.004 | 0.862 ±0.002 | 0.862 ±0.002 | 0.863 ±0.002 |
| **AANE** | 0.793 ±0.006 | 0.793 ±0.006 | 0.777 ±0.006 | 0.728 ±0.005 | 0.733 ±0.004 | 0.693 ±0.005 | **0.867** **±0.001** | **0.867** **±0.001** | **0.867** **±0.002** |
| **ASNE** | 0.831 ±0.003 | 0.830 ±0.003 | 0.812 ±0.004 | 0.713 ±0.004 | 0.718 ±0.004 | 0.677 ±0.004 | 0.846 ±0.002 | 0.846 ±0.002 | 0.843 ±0.002 |
| **BANE** | 0.807 ±0.005 | 0.807 ±0.005 | 0.787 ±0.005 | 0.707 ±0.003 | 0.713 ±0.003 | 0.670 ±0.004 | 0.823 ±0.002 | 0.823 ±0.002 | 0.822 ±0.002 |
| **TENE** | 0.829 ±0.005 | 0.829 ±0.005 | 0.815 ±0.004 | 0.664 ±0.004 | 0.681 ±0.003 | 0.611 ±0.002 | 0.842 ±0.001 | 0.842 ±0.001 | 0.843 ±0.002 |
| **AE** | 0.835 ±0.005 | 0.835 ±0.005 | 0.815 ±0.006 | 0.730 ±0.005 | 0.739 ±0.005 | 0.688 ±0.006 | 0.839 ±0.002 | 0.839 ±0.002 | 0.840 ±0.002 |
| **AE-EGO** | 0.835 ±0.005 | 0.835 ±0.006 | 0.816 ±0.005 | 0.729 ±0.004 | 0.739 ±0.005 | 0.690 ±0.007 | 0.840 ±0.002 | 0.840 ±0.003 | 0.839 ±0.002 |
| **MUSAE** | 0.848 ±0.004 | 0.848 ±0.004 | 0.832 ±0.005 | **0.737** **± 0.004** | **0.742** **±0.004** | **0.706** **±0.004** | 0.853 ±0.001 | 0.853 ±0.001 | 0.854 ±0.002 |
| **MUSAE-EGO** | **0.849** **±0.004** | **0.849** **±0.004** | **0.833** **±0.004** | 0.736 ±0.004 | 0.741 ±0.004 | 0.706 ±0.004 | 0.850 ±0.002 | 0.851 ±0.002 | 0.850 ±0.002 |

Table 6: Node classification test performance evaluated by weighted, micro and macro $F_1$ scores calculated from 10 seeded train-test splits. We included standard errors of the scores and used 80% of nodes for training / 20% of nodes for testing. Red numbers denote the best performing node embedding method and blue ones denote the best performing *supervised* graph neural network.

| | Datasets | | | | | | | | |
| --- | --- | --- | --- | --- | --- | --- | --- | --- | --- |
| | Facebook Page-Page | | | GitHub WebML | | | Twitch Portugal | | |
| | Weighted | Micro | Macro | Weighted | Micro | Macro | Weighted | Micro | Macro |
| **DeepWalk** | 0.861 ±0.001 | 0.863 ±0.001 | 0.848 ±0.001 | 0.852 ±0.001 | 0.858 ±0.001 | 0.801 ±0.002 | 0.650 ±0.008 | 0.672 ±0.007 | 0.594 ±0.009 |
| **LINE$_2$** | 0.874 ±0.001 | 0.875 ±0.001 | 0.862 ±0.001 | 0.852 ±0.001 | 0.858 ±0.001 | 0.800 ±0.002 | 0.636 ±0.006 | 0.670 ±0.005 | 0.571 ±0.005 |
| **Node2Vec** | 0.889 ±0.001 | 0.890 ±0.001 | 0.880 ±0.001 | 0.853 ±0.001 | 0.859 ±0.001 | 0.802 ±0.001 | 0.665 ±0.004 | 0.686 ±0.004 | 0.612 ±0.004 |
| **Walklets** | 0.886 ±0.001 | 0.887 ±0.001 | 0.875 ±0.001 | 0.854 ±0.001 | 0.860 ±0.001 | 0.804 ±0.002 | 0.652 ±0.006 | 0.671 ±0.006 | 0.599 ±0.005 |
| **TADW** | 0.760 ±0.002 | 0.765 ±0.002 | 0.740 ±0.003 | 0.650 ±0.001 | 0.748 ±0.001 | 0.528 ±0.007 | 0.459 ±0.001 | 0.659 ±0.005 | 0.406 ±0.003 |
| **AANE** | 0.793 ±0.001 | 0.796 ±0.001 | 0.775 ±0.001 | 0.848 ±0.001 | 0.856 ±0.001 | 0.794 ±0.002 | 0.636 ±0.006 | 0.661 ±0.006 | 0.577 ±0.006 |
| **ASNE** | 0.794 ±0.001 | 0.797 ±0.001 | 0.776 ±0.001 | 0.829 ±0.001 | 0.839 ±0.001 | 0.766 ±0.002 | **0.670 ±0.006** | **0.685 ±0.006** | **0.620 ±0.006** |
| **BANE** | 0.868 ±0.001 | 0.868 ±0.001 | 0.859 ±0.002 | 0.711 ±0.001 | 0.762 ±0.001 | 0.576 ±0.001 | 0.644 ±0.006 | 0.664 ±0.006 | 0.587 ±0.006 |
| **TENE** | 0.724 ±0.002 | 0.731 ±0.002 | 0.699 ±0.002 | 0.842 ±0.001 | 0.850 ±0.001 | 0.785 ±0.002 | 0.613 ±0.005 | 0.664 ±0.006 | 0.536 ±0.006 |
| **AE** | 0.887 ±0.001 | 0.888 ±0.001 | 0.879 ±0.001 | 0.858 ±0.001 | 0.863 ±0.001 | 0.807 ±0.001 | 0.653 ±0.005 | 0.672 ±0.004 | 0.598 ±0.006 |
| **AE-EGO** | **0.898 ±0.001** | **0.899 ±0.001** | **0.890 ±0.001** | 0.857 ±0.001 | 0.863 ±0.001 | 0.807 ±0.002 | 0.652 ±0.007 | 0.671 ±0.007 | 0.599 ±0.009 |
| **MUSAE** | 0.886 ±0.001 | 0.887 ±0.001 | 0.877 ±0.001 | **0.859 ±0.001** | **0.864 ±0.001** | **0.810 ±0.001** | 0.654 ±0.006 | 0.672 ±0.006 | 0.600 ±0.007 |
| **MUSAE-EGO** | 0.893 ±0.001 | 0.894 ±0.001 | 0.884 ±0.001 | **0.859 ±0.001** | **0.864 ±0.001** | **0.810 ±0.001** | 0.655 ±0.003 | 0.671 ±0.002 | 0.604 ±0.003 |
| **GCN** | 0.931 ±0.001 | 0.932 ±0.001 | 0.928 ±0.001 | 0.859 ±0.001 | 0.865 ±0.001 | 0.809 ±0.002 | 0.650 ±0.013 | 0.695 ±0.007 | 0.577 ±0.02 |
| **GraphSAGE** | 0.812 ±0.002 | 0.814 ±0.002 | 0.795 ±0.002 | 0.848 ±0.001 | 0.854 ±0.001 | 0.794 ±0.002 | 0.618 ±0.003 | 0.631 ±0.004 | 0.563 ±0.005 |
| **GAT** | 0.918 ±0.001 | 0.919 ±0.001 | 0.912 ±0.001 | 0.856 ±0.001 | 0.864 ±0.001 | 0.803 ±0.002 | 0.648 ±0.008 | 0.678 ±0.007 | 0.588 ±0.009 |
| **MixHop** | **0.940 ±0.001** | **0.941 ±0.002** | **0.937 ±0.001** | 0.847 ±0.000 | 0.85 ±0.001 | 0.800 ±0.001 | 0.626 ±0.003 | 0.630 ±0.004 | 0.576 ±0.003 |
| **ClusterGCN** | 0.937 ±0.001 | 0.937 ±0.001 | 0.934 ±0.001 | 0.855 ±0.001 | 0.859 ±0.001 | 0.807 ±0.001 | 0.647 ±0.004 | 0.654 ±0.004 | 0.602 ±0.005 |
| **APPNP** | 0.938 ±0.001 | 0.938 ±0.001 | 0.935 ±0.001 | **0.860 ±0.002** | **0.868 ±0.001** | **0.811 ±0.002** | **0.683 ±0.009** | **0.702 ±0.012** | **0.623 ±0.010** |
| **SGCONV** | 0.832 ±0.002 | 0.836 ±0.002 | 0.812 ±0.002 | 0.816 ±0.001 | 0.829 ±0.001 | 0.747 ±0.002 | 0.652 ±0.003 | 0.663 ±0.003 | 0.604 ±0.004 |

## H ADDITIONAL TRANSFER LEARNING RESULTS ON THE TWITCH GRAPHS

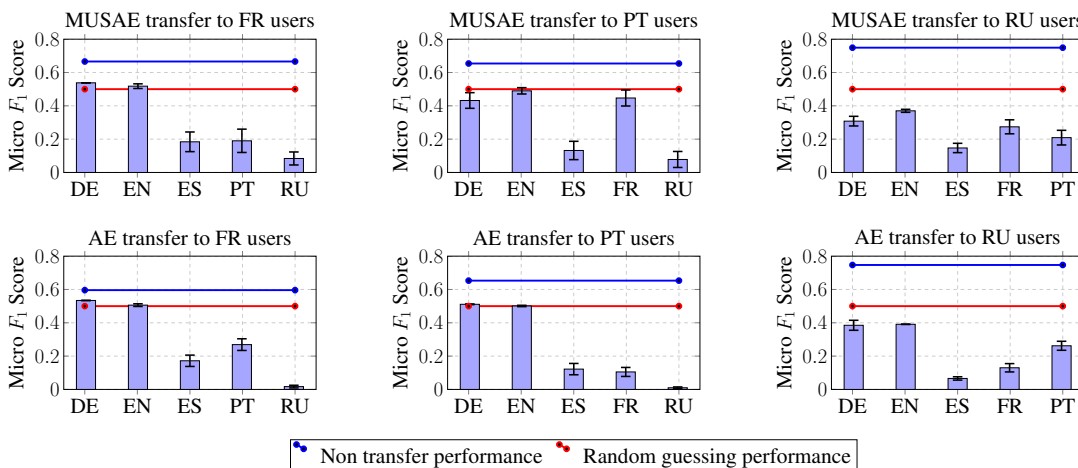

Figure 5: Mean micro $F_1$ scores and standard errors calculated from 10 transfer learning runs with *MUSAE* and *AE* on the Twitch graphs using France, Portugal and Russia as targets for the transfer. The blue reference line denotes the test performance on the target dataset in a non transfer learning scenario (standard hyperparameter settings and split ratio). The red reference line denotes the performance of random guesses.

## I  REGRESSION RESULTS ON WIKIPEDIA PAGE-PAGE GRAPHS

Table 7: Average test $R^2$ values and standard errors on the Wikipedia traffic prediction tasks. Red numbers denote the best results on each page-page network.

| Method | Datasets | | |
|---|---|---|---|
| | Wikipedia Chameleons | Wikipedia Crocodiles | Wikipedia Squirrels |
| DeepWalk | 0.375 ±0.004 | 0.553 ±0.001 | 0.170 ±0.002 |
| LINE$_2$ | 0.381 ±0.003 | 0.586 ±0.001 | 0.232 ±0.002 |
| Node2Vec | 0.414 ±0.003 | 0.574 ±0.001 | 0.174 ±0.002 |
| Walklets | 0.426 ±0.003 | 0.625 ±0.001 | 0.249 ±0.002 |
| TADW | 0.527 ±0.003 | 0.636 ±0.001 | 0.271 ±0.002 |
| AANE | 0.598 ±0.007 | 0.732 ±0.002 | 0.287 ±0.002 |
| ASNE | 0.440 ±0.009 | 0.572 ±0.003 | 0.229 ±0.005 |
| BANE | 0.464 ±0.003 | 0.617 ±0.001 | 0.168 ±0.002 |
| TENE | 0.494 ±0.02 | 0.701 ±0.003 | 0.321 ±0.007 |
| AE | 0.642 ±0.006 | 0.743 ±0.003 | 0.291 ±0.006 |
| AE-EGO | 0.644 ±0.009 | 0.732 ±0.002 | 0.283 ±0.006 |
| MUSAE | **0.658 ±0.004** | 0.736 ±0.003 | 0.338 ±0.007 |
| MUSAE-EGO | 0.653 ±0.011 | **0.747 ±0.003** | **0.354 ±0.009** |

## J  LINK PREDICTION RESULTS ON SOCIAL AND WEB NETWORKS

Table 8: Link prediction results - average AUC on the test set using attributed embeddings and logistic regression. We created 100 seeded splits (80% training - 20% test). Standard errors of AUC are included below. Red denotes the best performing embedding model considering both neighbourhood based and attributed methods. We used 4 element-wise operators to create features.

| Operator | Method | Datasets | | | | | |
|---|---|---|---|---|---|---|---|
| | | Facebook Page-Page | GitHub Web-ML | Twitch Spain | Twitch Germany | Wikipedia Chameleons | Wikipedia Crocodiles |
| Average | DeepWalk | 0.526 ±0.006 | 0.550 ±0.004 | 0.568 ±0.009 | 0.575 ±0.005 | 0.635 ±0.002 | 0.661 ±0.007 |
| | LINE$_2$ | 0.517 ±0.007 | 0.551 ±0.003 | 0.540 ±0.011 | 0.544 ±0.004 | 0.627 ±0.001 | 0.708 ±0.004 |
| | Node2Vec | 0.534 ±0.005 | 0.573 ±0.003 | 0.575 ±0.012 | 0.584 ±0.006 | 0.641 ±0.009 | 0.669 ±0.007 |
| | Walklets | 0.518 ±0.008 | 0.552 ±0.003 | 0.541 ±0.006 | 0.545 ±0.004 | 0.635 ±0.015 | 0.716 ±0.003 |
| Hadamard | DeepWalk | 0.981 ±0.001 | 0.799 ±0.001 | 0.781 ±0.002 | 0.750 ±0.003 | 0.974 ±0.002 | 0.966 ±0.001 |
| | LINE$_2$ | 0.979 ±0.001 | 0.899 ±0.001 | 0.843 ±0.003 | 0.755 ±0.001 | 0.939 ±0.003 | 0.938 ±0.001 |
| | Node2Vec | 0.982 ±0.001 | 0.822 ±0.001 | 0.810 ±0.005 | 0.780 ±0.003 | **0.979 ±0.001** | 0.973 ±0.001 |
| | Walklets | **0.984 ±0.001** | 0.925 ±0.001 | 0.873 ±0.003 | 0.819 ±0.001 | 0.966 ±0.004 | **0.980 ±0.001** |
| $l_1$ Norm | DeepWalk | 0.921 ±0.001 | 0.658 ±0.001 | 0.723 ±0.004 | 0.711 ±0.002 | 0.950 ±0.002 | 0.896 ±0.001 |
| | LINE$_2$ | 0.924 ±0.001 | 0.913 ±0.002 | 0.882 ±0.002 | 0.855 ±0.001 | 0.922 ±0.003 | 0.930 ±0.001 |
| | Node2Vec | 0.928 ±0.001 | 0.725 ±0.001 | 0.761 ±0.001 | 0.745 ±0.001 | 0.953 ±0.003 | 0.913 ±0.001 |
| | Walklets | 0.980 ±0.001 | 0.932 ±0.001 | **0.898 ±0.002** | 0.870 ±0.001 | 0.961 ±0.004 | 0.976 ±0.001 |
| $l_2$ Norm | DeepWalk | 0.922 ±0.001 | 0.663 ±0.001 | 0.731 ±0.004 | 0.717 ±0.002 | 0.951 ±0.002 | 0.899 ±0.002 |
| | LINE$_2$ | 0.924 ±0.001 | 0.910 ±0.001 | 0.880 ±0.002 | 0.855 ±0.001 | 0.925 ±0.003 | 0.936 ±0.001 |
| | Node2Vec | 0.929 ±0.002 | 0.731 ±0.001 | 0.768 ±0.007 | 0.750 ±0.001 | 0.954 ±0.002 | 0.920 ±0.001 |
| | Walklets | 0.981 ±0.001 | 0.930 ±0.001 | 0.897 ±0.002 | 0.870 ±0.002 | 0.960 ±0.005 | 0.978 ±0.001 |

Table 9: Link prediction results - average AUC on the test set using neighbourhood based embeddings and logistic regression. We created 100 seeded splits (80% training - 20% test). Standard errors of AUC are included below. Red denotes the best performing embedding model considering both neighbourhood based and attributed methods. We used 4 element-wise operators to create features.

| Operator | Method | Datasets | | | | | |
|---|---|---|---|---|---|---|---|
| | | Facebook Page-Page | GitHub Web-ML | Twitch Spain | Twitch Germany | Wikipedia Chameleons | Wikipedia Crocodiles |
| Average | TADW | 0.517 ±0.004 | 0.553 ±0.004 | 0.541 ±0.007 | 0.556 ±0.008 | 0.573 ±0.012 | 0.625 ±0.006 |
| | AANE | 0.523 ±0.003 | 0.539 ±0.003 | 0.536 ±0.001 | 0.554 ±0.004 | 0.552 ±0.013 | 0.577 ±0.005 |
| | ASNE | 0.547 ±0.005 | 0.596 ±0.002 | 0.562 ±0.007 | 0.579 ±0.005 | 0.650 ±0.005 | 0.723 ±0.004 |
| | BANE | 0.625 ±0.003 | 0.630 ±0.002 | 0.616 ±0.001 | 0.634 ±0.003 | 0.617 ±0.011 | 0.671 ±0.003 |
| | TENE | 0.547 ±0.004 | 0.515 ±0.004 | 0.532 ±0.011 | 0.555 ±0.006 | 0.555 ±0.011 | 0.644 ±0.003 |
| | AE | 0.572 ±0.006 | 0.719 ±0.002 | 0.679 ±0.006 | 0.723 ±0.003 | 0.669 ±0.007 | 0.834 ±0.002 |
| | MUSAE | 0.642 ±0.003 | 0.780 ±0.001 | 0.733 ±0.004 | 0.771 ±0.002 | 0.810 ±0.006 | 0.899 ±0.001 |
| | AE-EGO | 0.514 ±0.007 | 0.546 ±0.005 | 0.523 ±0.011 | 0.545 ±0.007 | 0.563 ±0.011 | 0.660 ±0.006 |
| | MUSAE-EGO | 0.511 ±0.005 | 0.542 ±0.003 | 0.533 ±0.005 | 0.546 ±0.008 | 0.622 ±0.008 | 0.699 ±0.003 |
| Hadamard | TADW | 0.973 ±0.001 | 0.915 ±0.001 | 0.886 ±0.003 | **0.884** ±0.001 | 0.964 ±0.002 | 0.967 ±0.001 |
| | AANE | 0.911 ±0.002 | 0.772 ±0.002 | 0.833 ±0.003 | 0.811 ±0.002 | 0.917 ±0.005 | 0.892 ±0.005 |
| | ASNE | 0.973 ±0.001 | 0.912 ±0.001 | 0.883 ±0.003 | 0.866 ±0.002 | 0.945 ±0.005 | 0.940 ±0.001 |
| | BANE | 0.653 ±0.002 | 0.664 ±0.003 | 0.659 ±0.009 | 0.816 ±0.002 | 0.578 ±0.014 | 0.738 ±0.002 |
| | TENE | 0.735 ±0.012 | 0.878 ±0.009 | 0.722 ±0.007 | 0.748 ±0.003 | 0.883 ±0.003 | 0.872 ±0.015 |
| | AE | 0.926 ±0.001 | 0.814 ±0.002 | 0.743 ±0.003 | 0.702 ±0.003 | 0.939 ±0.002 | 0.949 ±0.001 |
| | MUSAE | 0.945 ±0.001 | 0.917 ±0.002 | 0.871 ±0.005 | 0.863 ±0.002 | 0.950 ±0.005 | 0.968 ±0.001 |
| | AE-EGO | 0.928 ±0.001 | 0.786 ±0.002 | 0.727 ±0.006 | 0.687 ±0.003 | 0.935 ±0.002 | 0.939 ±0.001 |
| | MUSAE-EGO | 0.938 ±0.001 | 0.911 ±0.002 | 0.881 ±0.003 | 0.859 ±0.001 | 0.952 ±0.007 | 0.969 ±0.001 |
| $l_1$ Norm | TADW | 0.971 ±0.001 | 0.909 ±0.002 | 0.882 ±0.003 | 0.881 ±0.001 | 0.959 ±0.002 | 0.962 ±0.001 |
| | AANE | 0.866 ±0.002 | 0.720 ±0.001 | 0.771 ±0.004 | 0.768 ±0.001 | 0.944 ±0.002 | 0.913 ±0.001 |
| | ASNE | 0.815 ±0.002 | 0.866 ±0.001 | 0.836 ±0.002 | 0.849 ±0.001 | 0.869 ±0.001 | 0.874 ±0.001 |
| | BANE | 0.653 ±0.002 | 0.664 ±0.003 | 0.658 ±0.009 | 0.816 ±0.002 | 0.578 ±0.014 | 0.74 ±0.002 |
| | TENE | 0.940 ±0.001 | **0.942** ±0.001 | 0.857 ±0.004 | 0.837 ±0.001 | 0.945 ±0.003 | 0.927 ±0.001 |
| | AE | 0.968 ±0.001 | 0.889 ±0.001 | 0.871 ±0.001 | 0.870 ±0.002 | 0.955 ±0.002 | 0.952 ±0.002 |
| | MUSAE | 0.973 ±0.001 | 0.908 ±0.001 | 0.885 ±0.002 | 0.879 ±0.002 | 0.956 ±0.003 | 0.967 ±0.001 |
| | AE-EGO | 0.973 ±0.001 | 0.891 ±0.001 | 0.872 ±0.002 | 0.872 ±0.002 | 0.953 ±0.002 | 0.955 ±0.001 |
| | MUSAE-EGO | 0.977 ±0.001 | 0.911 ±0.001 | 0.891 ±0.002 | **0.884** ±0.002 | 0.955 ±0.003 | 0.963 ±0.001 |
| $l_2$ Norm | TADW | 0.972 ±0.001 | 0.913 ±0.001 | 0.883 ±0.003 | 0.879 ±0.001 | 0.961 ±0.002 | 0.964 ±0.001 |
| | AANE | 0.877 ±0.001 | 0.732 ±0.001 | 0.779 ±0.003 | 0.774 ±0.002 | 0.941 ±0.005 | 0.901 ±0.002 |
| | ASNE | 0.806 ±0.004 | 0.872 ±0.001 | 0.839 ±0.003 | 0.852 ±0.002 | 0.875 ±0.006 | 0.880 ±0.001 |
| | BANE | 0.653 ±0.002 | 0.664 ±0.003 | 0.659 ±0.009 | 0.816 ±0.0002 | 0.578 ±0.014 | 0.738 ±0.002 |
| | TENE | 0.893 ±0.001 | 0.884 ±0.016 | 0.826 ±0.009 | 0.797 ±0.005 | 0.930 ±0.003 | 0.863 ±0.013 |
| | AE | 0.968 ±0.001 | 0.881 ±0.001 | 0.872 ±0.001 | 0.867 ±0.002 | 0.954 ±0.003 | 0.953 ±0.001 |
| | MUSAE | 0.973 ±0.001 | 0.905 ±0.001 | 0.884 ±0.002 | 0.877 ±0.002 | 0.952 ±0.005 | 0.965 ±0.001 |
| | AE-EGO | 0.973 ±0.001 | 0.884 ±0.001 | 0.873 ±0.001 | 0.871 ±0.002 | 0.952 ±0.003 | 0.956 ±0.001 |
| | MUSAE-EGO | 0.977 ±0.001 | 0.907 ±0.001 | 0.891 ±0.003 | 0.881 ±0.002 | 0.951 ±0.006 | 0.961 ±0.001 |

