# OpenReview forum: "Multi-scale Attributed Node Embedding"
_ICLR.cc/2020/Conference — Reject_

### Official Review · AnonReviewer2 · 2019-10-23
**Official Blind Review #2**

**Rating:** 6

**Review:**

This paper proposed an attributed network embedding method by leveraging a node’s local distribution over attributes. The neighborhood attribute distribution of a node is considered in both a pooled and a multi-scale way. The multi-scale embedding approach considers the neighborhood nodes with different distance to the interested node distinctly, providing more flexibilities to the model. Then, the paper proved theoretically that the proposed embedding methods, both the pooled and multi-scale versions, can be equivalently written the factorization of a node-feature pointwise mutual information matrix.

The proposed embedding methods are standard. The key contribution of this paper comes from the theoretical part, which establishes the equivalence between the proposed embedding methods and matrix factorization. It looks interesting, although there are several similar works before, as mentioned in the paper. I don’t know how different your work is from the Qiu’s paper.

The experimental results are not convincing. The node classification is a very standard task in the performance evaluation of network embedding, but you put the results into the appendix. I examine the results anyway, and I found the performance gain is very limited, and on some datasets, the proposed methods even perform inferiorly.


**Experience Assessment:**

I have published one or two papers in this area.

**Review Assessment: Checking Correctness Of Derivations And Theory:**

I did not assess the derivations or theory.

**Review Assessment: Checking Correctness Of Experiments:**

I assessed the sensibility of the experiments.

**Review Assessment: Thoroughness In Paper Reading:**

I read the paper at least twice and used my best judgement in assessing the paper.

---

> ### Author Response · Authors · 2019-11-10
> **Author Response to Reviewer #2**
>
> "The proposed embedding methods are standard. The key contribution of this paper comes from the theoretical part, which establishes the equivalence between the proposed embedding methods and matrix factorization. It looks interesting, although there are several similar works before, as mentioned in the paper. I don’t know how different your work is from the Qiu’s paper."
>
> Response:
>
> The reviewer is correct that our work links into and builds on existing literature. In particular we extend the use of W2V-type algorithms to attributed networks (of which non attributed networks are a special case), which to our knowledge is novel. We also non-trivially extend the analysis that relates to those algorithms, e.g. that of Qiu. Note that we anticipate this to have important extensions as it unifies the embeddings of nodes with embeddings of node features, which are often words, and themselves can be embedded by W2V. Thus future work aims to unify attributed network embedding, where attributes are words, with word embeddings that can be learned from a vast independent corpus.
>
> "The experimental results are not convincing. The node classification is a very standard task in the performance evaluation of network embedding, but you put the results into the appendix. I examine the results anyway, and I found the performance gain is very limited, and on some datasets, the proposed methods even perform inferiorly."
>
> Response:
>
> We agree with the reviewer that node classification is a standard task and we are reviewing how the results might be better represented (Tables 6, 7 in Appendix G) within the paper. However, it is important to note that our algorithms learn fully unsupervised node/feature representations and the results for end-to-end supervised methods (e.g. GCN, GAT, GraphSAGE, ClusterGCN, APPNP) are included for reference as a proxy upper-bound. As expected, our algorithms do not perform as well as fully supervised methods, but are the best performing unsupervised methods overall. The fixed ratio train-test split experiments show our methods outperform the best performing unsupervised method by: 0.9% (Facebook Page-Page), 0.5% (Github), 0.8% (Cora), 0.8% (Citeseer) (which can be seen to be significant considering standard error). Where our models are outperformed (Pubmed, Twitch Portugal), it is by a different model in each case and our models are in the top tier. We note also that the regression results predicting Wikipedia traffic (Appendix I), a related task to node classification, show a material benefit of our models over other unsupervised methods varies (1.5% - 6.0% $R^2$). We have highlighted that Table 7 includes fully supervised methods for greater clarity.

---

### Official Review · AnonReviewer3 · 2019-10-23
**Official Blind Review #3**

**Rating:** 6

**Review:**

This manuscript introduces embedding algorithms that consider attribute distribution. To address the multi-scale attribute information, the multi-scale version of AE is derived (MUSAE). Then the proposed algorithms are proven theoretically to implicitly factorize the PMI matrix, which enhance their interpretability. The experiments are conducted on various scenarios including node classification, transfer learning, regression and link prediction. showing the quality of learned embeddings. The results show the benefits of multi-scaling and several conclusions are drawn.
Following are some review’s questions:
1. In MUSAE, what is the intention that the tuples are added to different sub-corpus for source and target nodes? Besides, the D_r should be a corpus rather sub-corpus.
2. In 5.2, I’m not quite understand what do you mean by ‘vanilla MUSAE and AE are inductive and can easily map nodes to the embedding space if the attributes across the source and target graph are shared’.

**Experience Assessment:**

I do not know much about this area.

**Review Assessment: Checking Correctness Of Derivations And Theory:**

I assessed the sensibility of the derivations and theory.

**Review Assessment: Checking Correctness Of Experiments:**

I assessed the sensibility of the experiments.

**Review Assessment: Thoroughness In Paper Reading:**

I read the paper at least twice and used my best judgement in assessing the paper.

---

> ### Author Response · Authors · 2019-11-10
> **Author Response to Reviewer #3**
>
> 1. In MUSAE, what is the intention that the tuples are added to different sub-corpus for source and target nodes? Besides, the D_r should be a corpus rather sub-corpus.
>
> Response:
>
> In AE, as in many W2V-inspired network embedding models, one central corpus $\mathcal{D}$ is constructed containing information from node pairs over different proximities, i.e. the information is “pooled”. However, one intuitively expects the relationship between nodes to vary with proximity ($r$), hence rather than pooling all information together, MUSAE partitions it between “sub-corpora” $\mathcal{D}_r$. This gives down-stream tasks access to unpooled, and thus more fine-grained information, with the potential draw-back that, since more statistical information (i.e. for each proximity $r$) is estimated from the same dataset, those estimates are likely to contain more variance.
>
> 2. In 5.2, I’m not quite understand what do you mean by ‘vanilla MUSAE and AE are inductive and can easily map nodes to the embedding space if the attributes across the source and target graph are shared’.
>
> Response:
>
> MUSAE and AE (as opposed to their EGO counterparts) learn node embeddings as a function of the features around them, as opposed to the nodes around them. If two networks share a common feature domain, then feature representations learned on one graph can be used to generate node representations on the other (on the assumption that nodes exhibit features for the same underlying latent reasons). However, since neighbourhood-based methods learn representations as a function of their node neighbourhood, nodes from separate graphs cannot be mapped to the embedding space using the same embedding function, a strong limitation compared to our method.

---

### Official Review · AnonReviewer1 · 2019-10-24
**Official Blind Review #1**

**Rating:** 3

**Review:**

This paper introduces Skip-gram style embedding algorithms that consider attribute distributions over local neighborhoods. Algorithm 1 and 2 shows that in fact they propagate randomly selected node features to neighbors. The reviewer doesn’t think this random-walk way for selecting node feature is appropriate.  Node features describe node content. The features of neighboring nodes may complement each other. However, there is no benefit to select random features and then propagate, given that there already many approaches smartly combining node content in neighborhood.

The proof part follows Qiu et al (2018). But it is unclear, why c^{−1}A is the stationary joint distribution over consecutive nodes p(wj , wj+1).  and c^{−1}DF describes the stationary joint distribution p(f,wj) over nodes and features. There needs more explanation.

The Remark 1 and 2 discuss the case AE with F=I_|V|,  which is in fact the case when there is no node attributes. In this case, the AE process naturally goes back to plain network embedding, where DeepWalk and WalkLets are proposed for. Therefore, these remarks are done by Qiu et al (2018) already, not make no much new contribution here.

Figure 2 shows that the presented two approaches just slightly better or equivalent, or sometimes worse than baseline methods. As mentioned earlier, it is not beneficial to randomly select features to propagate.

The experiments presented in Section 5.2 evaluate whether the learned embedding can be used for label inference in a different graph. But it is unknown how success the transferring is. There is no F1-score of a solution that does embedding of the target network itself independently, and then classify the target network nodes.


**Experience Assessment:**

I have published in this field for several years.

**Review Assessment: Checking Correctness Of Derivations And Theory:**

I carefully checked the derivations and theory.

**Review Assessment: Checking Correctness Of Experiments:**

I carefully checked the experiments.

**Review Assessment: Thoroughness In Paper Reading:**

I read the paper thoroughly.

---

> ### Author Response · Authors · 2019-11-10
> **Author Response to Reviewer #1 - Part #1**
>
> "This paper introduces Skip-gram style embedding algorithms that consider attribute distributions over local neighborhoods. Algorithm 1 and 2 shows that in fact they propagate randomly selected node features to neighbors. The reviewer doesn’t think this random-walk way for selecting node feature is appropriate.  Node features describe node content. The features of neighboring nodes may complement each other. However, there is no benefit to select random features and then propagate, given that there already many approaches smartly combining node content in neighborhood."
>
> Response:
>
> We fully agree with the reviewer that node features describe node context. However, we would like to clarify that in our algorithm node features are not propagated randomly. As in many graph embedding models (e.g. Deepwalk, Walklets, Node2Vec), stochastic random walks are used to propagate information “in expectation”, since to compute all possible paths, equivalent to breadth-first search, is prohibitive for large graphs. However, ALL node pairs (up to a proximity $t$) are extracted from those paths and ALL features of each node pair propagated. We make this more clear in the paper by stating “for all $f \in \mathcal{F}$” (e.g. Alg 1, line 6).
>
> "The proof part follows Qiu et al (2018). But it is unclear, why $c^{−1}\textbf{A}$ is the stationary joint distribution over consecutive nodes $p(w_j , w_{j+1})$.  and $c^{−1}\textbf{D}\textbf{F}$ describes the stationary joint distribution $p(f,w_j)$ over nodes and features. There needs more explanation. "
>
> Response:
>
> We agree with the reviewer that this is not well explained (due to space) and will add the explanation below to the appendix. Note: we do not use these descriptions, but include them to aide intuition of the results/proofs.
> Since $(1/c)\textbf{D}$ is a diagonal matrix of the stationary distribution over nodes $p(u)$ and $\textbf{P}=\textbf{D}^{-1}\textbf{A}$ is the transition matrix of probabilities $p(v|u)$, then:
>   $$(1/c)\textbf{A} = (1/c) \times [\textbf{D}\times \textbf{D}^{-1}] \times \textbf{A} = (1/c)\textbf{D} \times \textbf{D}^{-1}\textbf{A} = diag[p(u)] \times [p(v|u)] = [p(u,v)],$$
> a matrix of joint distributions of being in node $u$ and transitioning to node $v$. Since $p(u)$ is stationary, this joint distribution is the stationary distribution over consecutive node pairs.
>
> Similarly, $(1/c)\textbf{D} \textbf{F} = (1/c)\textbf{D} \times \textbf{F} = diag[p(u)] \times [p(f|u)] = [p(u,f)]$, a matrix of joint distributions of being in node $u$ and observing feature $f$, which is again a stationary distribution as in expectation it will remain unchanged between time steps.
>
> "The Remark 1 and 2 discuss the case AE with $\textbf{F}=\textbf{I}_{|V|}$,  which is in fact the case when there is no node attributes. In this case, the AE process naturally goes back to plain network embedding, where DeepWalk and WalkLets are proposed for. Therefore, these remarks are done by Qiu et al (2018) already, not make no much new contribution here. "
>
> Response:
>
> The reviewer is correct that when the feature matrix $\textbf{F}$ is the identity matrix, attributed network embedding becomes equivalent to unattributed network embedding. The work of Qiu relates to unattributed embedding only and hence our Remarks 1 and 2, which situate our method amongst others in the literature, also connect it to the work of Qiu, (some of) which we generalise to attributed networks.

---

> ### Author Response · Authors · 2019-11-10
> **Author Response to Reviewer #1 - Part #2**
>
> "Figure 2 shows that the presented two approaches just slightly better or equivalent, or sometimes worse than baseline methods. As mentioned earlier, it is not beneficial to randomly select features to propagate. "
>
> Response:
>
> We agree with the reviewer that Figure 2 does not represent the results sufficiently clearly due to varying y-axis scales of the subplots. The first plot (Facebook dataset) shows that all 4 proposed algorithms materially outperform (by 10-20%) all other models except BANE, which is outperformed but less so. The second plot (Github dataset) shows similar significant outperformance of other models, but here TENE is the only close competitor. The third plot (Twitch dataset) shows much closer results for all models (ASNE shows slight improvement by c 2%, however, it is significantly outperformed on other datasets). We have updated the figure such that all subplots share the same y-axis. Whilst this loses detail on the third plot, more importantly, it allows performance to be clearly compared between datasets, showing that the Twitch dataset is more difficult for all models. Note that features are not randomly propagated as explained above.
>
> We also emphasize that similar results are shown for fixed ratio train-test splits on these datasets (Appx G, Table 7), which also compare to end-to-end supervised methods. Results for citation graphs (Appx G, Table 6) again show that no other unsupervised method consistently outperforms our proposed models (e.g. AANE outperforms on Pubmed, but is outperformed on other datasets, particularly so on Cora).
>
> "The experiments presented in Section 5.2 evaluate whether the learned embedding can be used for label inference in a different graph. But it is unknown how success the transferring is. There is no F1-score of a solution that does embedding of the target network itself independently, and then classify the target network nodes."
>
> Response:
>
> We agree with the reviewer’s comment. We have added two baselines.  The first one is to make clear the relative benefit of transfer learning by independently learning features on the target network and a downstream logistic regression model using those (high baseline). Under the assumption of random guessing the labels with equal probability the micro F-1 score is 0.5 - we decided to use this as a low baseline. Note that it is possible that the high baseline could be outperformed by transfer learning, subject to the relative amount of data for each network. We added the low and high baselines to Figure 3.

---

### Author Response · Authors · 2019-11-11
**Summary of changes made in the revised version**

We thank all reviewers for their valuable feedback. We have responded to most comments individually. The paper has been updated to include:

- Low and high baseline for transfer learning.

---

### Decision · Program_Chairs · 2019-12-19

**Decision:**

Reject

**Comment:**

This paper constitutes interesting progress on an important topic; the reviewers identify certain improvements and directions for future work, and I urge the authors to continue to develop refinements and extensions.